# Symbolic and non-symbolic numbers differently affect center identification in a number-line bisection task

Annamaria Porru[1]*, Lucia Ronconi[2], Daniela Lucangeli[1], Lucia Regolin[3], Silvia Benavides-Varela[1,4], Rosa Rugani[3]

**1** Department of Developmental Psychology and Socialization, University of Padua, Padua, Italy, **2** Computer and Statistical Services, Multifunctional Pole of Psychology, University of Padua, Padua, Italy, **3** Department of General Psychology, University of Padova, Padua, Italy, **4** Padova Neuroscience Center, University of Padua, Padua, Italy

☉ These authors contributed equally to this work.
* annamaria.porru@unipd.it

## Abstract

Numerical and spatial representations are intertwined as in the Mental Number Line, where smaller numbers are on the left and larger numbers on the right. This relationship has been repeatedly demonstrated with various experimental approaches, such as the line bisection task. Spatial accuracy appears to be systematically distorted leftward for smaller digits by elaboration of spatial codes during number processing. Other studies have investigated perceptual and visuo-spatial attention bias using the digit line bisection task, suggesting that these effects may be related to a cognitive illusion in which the reference numbers project their values onto the straight line, creating an illusory lateral disparity. On the other hand, both dot arrays (non-symbolic stimuli) and arabic numbers (symbolic stimuli) demonstrate a privileged relation between spatial and numerical elaboration. The bias toward the larger numerosity flanker was attributed to a length illusion. There is, however, no consensus regarding whether physical features and symbolic and non-symbolic numerical representations exert the same influence over spatial ones. In the present study, we carried out a series of 4 Experiments to provide further evidence for a better understanding of the nature of this differential influence. All experiments presented the numbers in both symbolic and non-symbolic formats. In Experiment 1, the numbers "2-8" were presented in a variety of left-right orientations. In Experiment 2, the flankers were identical, "2-2" or "8-8", and symmetrically displaced with respect to the line. In Experiment 3, we employed asymmetrically distributed eight dots, or font sizes in "8-8" numerals, to create a perceptual imbalance. In Experiment 4, we replicated the manipulation used in Experiment 3, but with two dots and "2-2" numerals. The Non-Symbolic format induced stronger leftward biases, particularly when the larger numerosity (Experiment 1) or the denser stimuli near the line (Experiments 3 and 4)

**Data availability statement:** The datasets, script and stimuli are available on OSF at link: https://osf.io/hr8sz/?view_only=bc-629be803724e819df7c31e5ddbd213.

**Funding:** This work is funded by the European Union – Next Generation EU Prot. PRIN 2022 PRIN -202254RHRT to R.R., and PRIN 2022 PNRR - P2022TKY7B to S.B.-V. The funders had no role in study design, data collection and analysis, decision to publish, or preparation of the manuscript.

**Competing interests:** The authors have declared that no competing interests exist.

were on the left, while no bias emerged when flankers were numerically equivalent and symmetrical (Experiment 2). The left bias may result from a tendency to estimate the influence of stimulus perception associated with participants' scanning direction, similar to the direction of pseudoneglect. Conversely, the Symbolic format induced mostly right bias, possibly due to left-lateralized processing and a tendency to use a common strategy involving scanning from left to right. Altogether our data support the view that abstract numbers and non-symbolic magnitude affect perceptual and attentional biases, yet in distinctive ways.

## Introduction

Humans organize numerical information in space according to a specific left-to-right orientation: the so-called Mental Number Line [MNL; 1]. The evidence for the orientation of the mental number line representation first comes from the SNARC effect (Spatial-Numerical Association of Response Codes), which has been interpreted on the basis of a spatial-congruency between the response side (the left and right sides of egocentric space), and the relative position of the represented numerical magnitude on an oriented mental number line (the left-space/small numbers and the right space/large numbers) [2,3]. Spatial-numerical association (SNA) can be observed in a variety of tasks [4], age groups (newborns: [5,6], infants [7,8], preschoolers [9], and species [10–12]) and has been conceptualized as an evolutionarily prerequisite for the emergence of the mental number line.

The line bisection task, historically used as a diagnostic tool for assessing spatial attention deficits in patients with neurological disorders [13,14], has been adapted in several studies to investigate SNA in healthy participants. To this aim, the line has been typically constructed with strings of digits [15], or as a line with numerical flankers to investigate how number processing influences spatial biases [i.e., 16–20]. The results generally indicate a tendency to bisect towards the larger number [15,16,21], or larger area; in the case of non-symbolic displays [22,23]. Whether biases in the line bisection task result from perceptual [i.e., 24], attentional [i.e., 16] or conceptual factors [i.e., 7] has been the subject of intense debate, and remains a matter of controversy [22,23].

The influence of number processing over spatial judgments was first evident in a landmark study by Martin Fischer [15]. The author designed a paper-and-pencil task requiring neurologically healthy participants to mark the midpoint of strings composed of Arabic numerals. Strings formed by 1's or 2's induced a left bias, while strings of 8's or 9's induced a right bias. This phenomenon was explained in terms of the mental number line hypothesis, namely an automatic association of number magnitudes with spatial representations. Such a bias was replicated also when the strings were replaced with lines flanked by numerals (i.e., 1–2, 2–2, 8–9, 9–8). In all the conditions the midpoint was misjudged toward the flanker representing the larger magnitude. Subsequent studies demonstrated that the bias is present, and even more pronounced, with number words (e.g., 'DEUX', 'NEUF') than numerals (e.g., 2, 9) [17], and with larger numerical intervals (e.g., 1–8, 2–9) than smaller ones (e.g., 1–2, 8–9) [18].

Further, research integrating a line bisection task with numerical information in neglect patients showed a right bias when lines were flanked by large numerals (9–9) and a left bias when the flankers were small (1–1) [19]. Altogether these studies provide evidence that processing numerical information systematically biases spatial performance in the line bisection task, further supporting the hypothesis of an automatic association between numbers and space [15].

DeHevia and colleagues [16] subsequently performed a series of studies to disentangle the factors that influence these biases. The authors found, contrary to Fischer's [15] findings, that the magnitude of the numbers does not modulate the performance in the task. Instead, other properties such as spatial reference frames (i.e., whether the line was placed in the left, right or central position of the page; Exp 1), proximity of the flankers (Exp 3), attentional cueing (Exp 4) and pseudoneglect (Exp 2) - a subtle leftward deviation of the midpoint of a horizontal segment, commonly observed in neurologically unimpaired subjects [25] were found to be most influential. According to the authors, deviations in the line bisection may be related to a cognitive illusion, in which the numerical flankers create an illusory spatial extension of the line [16].

Does numerical information exert an influence on the mental representation of spatial extent independently of cultural and maturational factors such as symbolic knowledge and age? DeHevia & Spelke [7] investigated this question by introducing a non-symbolic flanker task. They directly examined the hypothesis that the mental number line arises from an inherent tendency of the human mind to associate spatial and numerical representations, rather than being a product of cultural artifacts.

In accordance with their hypothesis, results showed that the non-symbolic format (arrays of 2 or 9 dots) systematically distorted the localization of the midpoint of a horizontal line in adults as well as in young children. Moreover, because adults showed a consistent bias toward the more numerous array in both the symbolic and the non-symbolic format [7], this was taken as evidence that the underlying numerical representation is common to both formats reflecting an abstract mapping between numerical magnitude and horizontal space [7; for consistent results see 26–28].

Nevertheless, contrasting evidence has been also documented. An increasing number of findings support distinct mechanisms for the elaboration of symbolic and non-symbolic number representations [29–36], questioning the conclusion that the spatial biases observed with either dot-arrays or arabic numbers could be comparable. Crucially, in non-symbolic displays, weighting different visual cues could hinder or influence the extraction of numerosity information, affecting automatic associations between numbers and space and therefore the direction of the biases. For example, Gebuis and Gevers [22] showed that participants, during a non-symbolic line bisection task, identified the midpoint more accurately when larger numerosity had the smallest total area. Additionally, when flankers were numerically identical but varied in physical information (i.e., area, surface) participants consistently bisected toward the larger area [23], highlighting the relevance of visuo-perceptual cues in numerosity processing. In another study, when symbolic and non-symbolic numbers were simultaneously displayed (with digits presented instead of dots, in a dice-like pattern), the automatic association between numbers and space emerged exclusively for the symbolic forms, thereby reinforcing the concept of distinct representations for the two formats [29].

At present, this mixed evidence leaves open the question of whether symbolic and non-symbolic formats induce comparable effects on the processing of spatial extension in line bisection tasks. A further area of ambiguity pertains to the determination of the factors influencing the bias in numerical line bisection tasks, with uncertainty surrounding whether these factors are governed by the conceptual mental number line or by attentional or perceptual factors. Here, we address this question by specifically exploring whether unbalancing different perceptual properties of the flankers in the two formats influences the core aspects of the automatic association between numbers and space to the same extent.

In Experiment 1, which was inspired by the study of De Hevia & Spelke [7], lines were flanked by different numbers (2–8 or 8–2) that were presented as either Arabic digits (Symbolic format) or dots (Non-Symbolic format). The experiment aimed to reproduce the original findings; therefore, methodological choices were guided by their implementations (see methods section). We expected to observe a bisection bias towards the larger quantity, regardless of the type of stimulus, consistent with the proposal that numerical-space associations derive from an abstract mental representation of numbers.

In the following experiments, we further explored the hypothesis of a common effect across Symbolic and Non-Symbolic representations by pairing deep/conceptual features of the flankers (i.e., numerosity) while manipulating shallow/perceptual ones (i.e., proximity to the line, size, grouping). In Experiment 2 the flankers, independently of the format, were physically the same and symmetrically positioned relative to the line. We expected either an accurate middle identification due to the symmetry of the display and equal influence of the symbolic and non-symbolic flankers on the line, or alternatively, a leftward bias, consistent with the manifestation of the pseudoneglect phenomenon [25]. In Experiments 3 and 4, although the numerical and associated non-numerical information (i.e., subtended area, aggregate surface area, etc.) was the same in both flankers, their perceptual saliency differed. In the Symbolic format, one of the flankers was printed in a larger font size with respect to the other. In the Non-Symbolic format, one of the flankers displayed a large number of dots grouped in the proximity of the line whereas the other one displayed them on the opposite side of the line. These manipulations allowed a direct test of the role of perceptual/attentional factors for bisection judgment in both formats. If these factors play a significant role in producing the bisection bias, then subjects may tend to shift the bisection marks towards the most salient stimuli, independently of stimuli format. Alternatively, if the abstract numerical information extracted from the array is crucial, and the format and perceptual magnitude are irrelevant, no bias is expected.

## Method

### Participants

An a priori power analysis was conducted using G*Power 3 icon (Version 3.1.9.6) to determine the appropriate sample size for our study. Based on the study by De Hevia and Spelke [7], we set the following parameters: power = .80, α = .05, Effect size Cohen's f = .25. The power analysis was originally designed for repeated-measures within-subjects ANOVA. However, the actual analysis conducted in this study employed Linear Mixed Effects Models (LMMs), which better account for individual differences and the complex structure of the data. The parameters selected for the ANOVA offer a general estimation. Given that LMMs frequently provide enhanced flexibility and power in detecting effects due to their capacity to model random effects, we consider the power analysis conducted here to be a conservative estimation.

According to the results of the power analysis, the total sample size needed is 34 participants for each experiment. This study comprised 4 different groups. Thus, in total 136 participants were enrolled, 34 in each group. Ten participants were excluded during the analyses because their mean bias was more than 2 SD above the group mean. Three participants were replaced in each of Experiments 1, 2, and 3, and one in Experiment 4. For this reason, we recruited an additional 10 more participants to reach the planned sample size. All participants were Italian students at the University of Padua, Italy. The mean age was 19.89 years (SD = 1.35). All participants were right-handed (based on self-reports) and had normal or corrected-to-normal sight. All participants provided signed written informed consent prior to the study and were naïve about the hypotheses of the experiment. Participants were recruited between May 28 and July 28, 2023. The numerical bias phenomenon was explained at the end of the experiment.

This study was approved by the Ethics Committee of the University of Padua (protocol number: 5326) and performed in accordance with the principles expressed in the 1964 Declaration of Helsinki.

### Stimuli and materials

Each participant received a paper block with 132 horizontally oriented sheets (297mm x 105mm). The stimuli were printed in black on white paper. The sheets were 12 gsm (grams per square meter) thick, to counteract any interference effects due to the transparency of the sheets. Each sheet depicted a horizontal black line printed in the center. The line was 1 mm wide, the length was either 60 or 80 mm and it was counterbalanced across trials. The presentation of each line individually enables the entire page as spatial reference, and reduces other potential referential effects from stimuli printed in the sheet, or from previous responses.

In the Non-Symbolic format, the stimuli used in Experiments 1 and 2 were generated using the GeNEsIS software [37]. The total perimeter was controlled to 7.5 cm for both stimuli. Sixteen different arrays of dots were generated to control and test for configural effects (see OSF https://osf.io/hr8sz/?view_only=bc629be803724e819df7c31e5ddbd213 for major info on each experimental task). The stimuli were presented within a visible square of 5cmx5cm.

In the Symbolic format, for Experiments 1 and 2, the Arabic digits were printed in a black Arial font, size 30 points (5mm wide and 7,5mm high in the printed version). For Experiment 3 and 4, we used two sizes: 30 points and 60 points (10mm wide and 15mm high). The stimuli were presented within an invisible square of 5cmx5cm that served as the framework for the accurate location of the digit across trials and conditions. The digit was vertically centered with respect to the square and horizontally aligned towards the line. The line was centered vertically, with the digits positioned equidistant from both ends.

In *Experiment 1,* the actual flanker stimuli used were "2" on the left and "8" on the right for the Small-left orientation and *viceversa* for the Large-left orientation (see Fig 1a).

In *Experiment 2,* the two flanker stimuli were identical. The stimuli in the Non-Symbolic format were symmetrical and specular.

In *Experiment 3,* the two flanker stimuli were numerically identical ("8-8"), but in an effort to induce a perceptual imbalance while maintaining the same numerosity and configural information, the stimuli were disproportionately distributed. Specifically, in the Non-Symbolic format, a virtual -not visible- vertical line divided the outer frame in two halves with 6 dots in one half and 2 dots in the other half. In the Small-left orientation, the half with two dots was closer to the left side of the line whereas in the Large-left orientation the half with six dots was closer to the left side of the line.

In *Experiment 4*, like in Experiment 3, the flanker stimuli were identical ("2-2") but unevenly distributed within the stimulus square. More precisely, in the Non-Symbolic format, the flankers were positioned such that the two dots appeared in one half of the square, while the other half was left empty. In the Small-left orientation, the empty half was closer to the left side of the line, whereas in the Large-left orientation the half depicting two dots was displayed to the left side of the line (see Fig 1a).

## Procedure

The present study adapted the line bisection task previously used by De Hevia et al, [16] and by De Hevia and Spelke, [7]. Before starting the experiment, participants received general instructions about the task. They were informed that on each trial they will be presented with one horizontal line printed on a paper sheet and received instructions to mark with a pencil, rapidly and accurately, the center of each line. The paper and pencil modality was the most conventional but also comparable to previous research.

The participants were also told to ignore the flanking stimuli. Stimuli were presented one at a time, aligned with reference to the mid-sagittal plane of the body. The distance between the sheet and the participant was approximately 30 cm. Once the stimulus had been bisected, participants were asked to continue to the next trial.

## Design

The experiments consisted of a test phase of 128 trials preceded by a baseline phase of 4 trials. In each experiment, the baseline phase was always presented at the beginning of the task. While in the baseline only the line was displayed, in the test phase the line was presented with flankers on each edge. Flankers in the test phase had two formats, they were either digits (Symbolic format) or dot arrays (Non-Symbolic format). Furthermore, when asymmetrical, flankers could be arranged so that either the most prominent (Large-left) or the least prominent (Small-left) element appears on the left side of the line. This resulted in four conditions (2 formats x 2 orientations) (see Fig 1a). When flankers were symmetrical (Experiment 2), they lacked the orientation feature. We arbitrarily designated "2-2" as the Small-left orientation and "8-8" as the Large-left orientation, to keep the statistical models constant across Experiments.

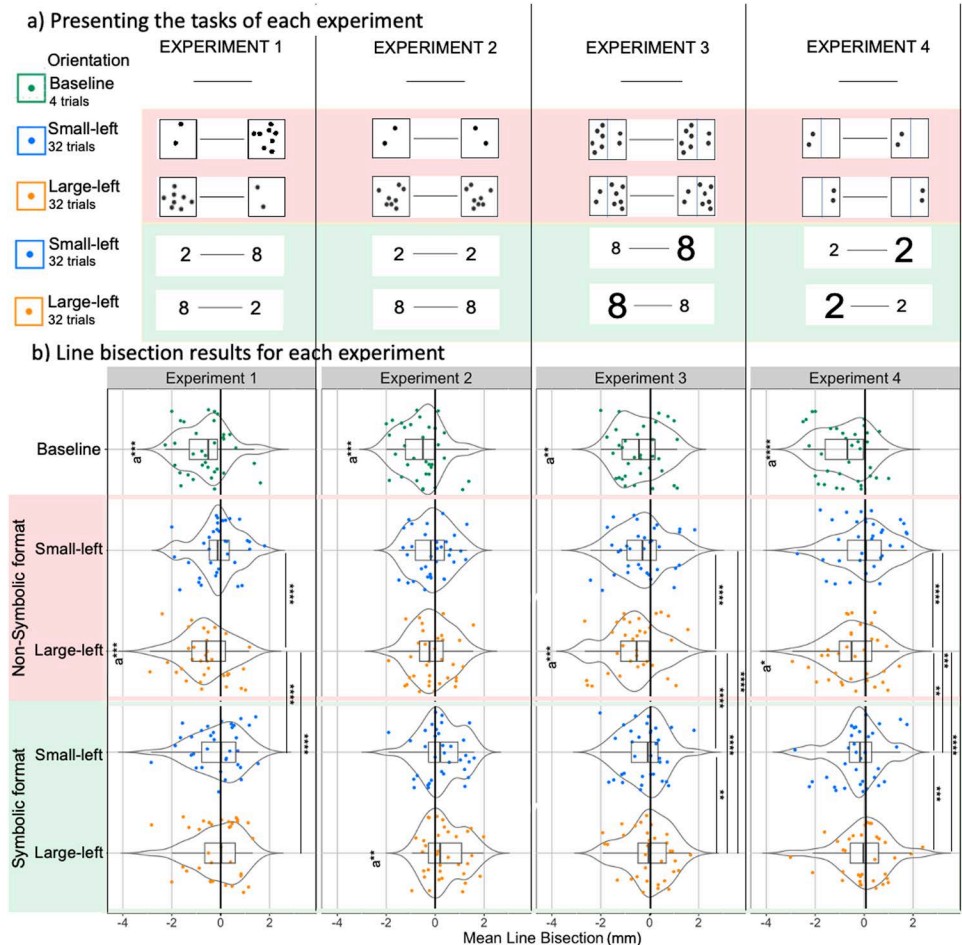

**Fig 1. Experimental stimuli and Results of line bisection with Post-hoc analyses in the Linear Mixed-Effects model.** In Experiment 1, the flankers were "2-8" or "8-2". In Experiment 2, the flankers were identical, "2-2" or "8-8", and symmetrically displaced with respect to the line. In Experiment 3, the stimuli (numerosity 8) were asymmetrically distributed to create a perceptual imbalance. In Experiment 4, we replicated the manipulation used in Experiment 3, but with the numerosity 2. All experiments presented the numbers in both symbolic and non-symbolic formats. a) Shows the description of the tasks in the different experiments, divided into two blocks: Non-Symbolic (in pink) and Symbolic (in green), with the two orientations Small-Left and Large-Left. b) Shows the results of line bisection with post-hoc analyses in the Linear Mixed-Effects model. The x-axis shows the distance in mm from the centre, for both the Non-Symbolic and Symbolic blocks, each divided into the two orientations Small-Left and Large-Left. The box plots show the mean, standard deviation (ds) and standard error (se), while the violin plots show the distribution of the data with each dot corresponding to one participant. *Note* (* $p < .05$, ** $p < .01$, *** $p < .001$, **** $< .0001$, a = One Sample t-test against 0).

All the trials of a given condition were presented in 4 blocks with 32 trials each. The order of trials within each block was random. Blocks in the Symbolic and Non-Symbolic format were presented in alternation. The order of presentation was randomized across participants so that half of them completed the Non-Symbolic format first and viceversa for the other half.

## Data coding

Bisection marks were measured to the nearest mm, rounded to the nearest half unit, using a ruler at the point where the bisection mark intersected the line, as in De Hevia and Spelke [7]. Our accuracy using a ruler is approximately ±0.5 mm. Deviations from the objective center towards the left were scored as a negative value, deviations towards the right as a

positive value. Participants were excluded from the analyses because their mean bias was more than 2 SD above the group mean.

## Data analysis

In all the studies, Linear Mixed Effects Models (LMMs) were applied to investigate the extent to which participants' line bisection was influenced by format (Symbolic *vs.* Non-Symbolic) and orientation (Small-left *vs.* Large-left). The dependent variable was the mean deviation in mm of each mark with respect to the objective center. The initial models included the main effects of format and orientation and interactions between these factors.

In addition, to control for the repeated measurement in the LMMs, participants were treated as a random factor in all the models. These analyses were performed by fitting a LMMs using the lme4 and lmerTest package [Version 1.1; 38,39] in RStudio [Version 2023.03.1 Build 446; R Development Core Team, 2016]. Post hoc tests were conducted by using the emmeans function provided by the emmeans package [40] and the Tukey Method was used for pairwise comparisons while controlling for all family error rates of 4 estimates. The significance level of statistical tests was predetermined at p = 0.05. Moreover, a one-sample t-test was conducted independently of the LMM analysis to evaluate whether the mean deviation in each condition was significantly different from the objective center (zero). In order to integrate the results of the LMMs, analysis of variance (ANOVA) was performed on the subjects' means, including line length, orientation and format as a factor. In addition, a t-test was performed to compare baseline with format and orientation combined, based on the subjects' means. The results of the t-test (S6 Table) and the ANOVA (S7 File) are reported in the Supporting Information.

## Results

### Experiment 1

The results showed that flanker format (Symbolic *vs.* Non-Symbolic) had a significant effect (β = 0.46, SE = 0.05, t (4303.00762) = 8.19, p < 0.001), since larger deviations were registered in the Non-Symbolic format (M = -0.35 mm; DS = 1.49 mm) than in the Symbolic format (M = -0.15 mm; DS = 1.62 mm) (refer to Table A and Table B in S5 Table for comprehensive mean value). Orientations (Small-left *vs.* Large-left) also had a significant effect on bisection (β = 0.51, SE = 0.055, t(4303.00509) = 9.14, p < 0.001), with larger deviations in the Large-left orientation trials (M = -0.38 mm; DS = 1.59 mm) compared to the Small-left orientation trials (M = -0.12 mm; DS = 1.52 mm).

Additionally, there was a significant interaction between flanker orientation and format (β = -0.50, SE = 0.07, t (4303.00343) = -6.41, p < 0.001) (refer to Fig 1b). Random intercept variance for participants was σ² = 1.68 (SD = .74). The variance explained by fixed effects was R²m = .02, while the total explained variance was R²c = .32. The intraclass correlation coefficient (ICC) was .31, indicating that 31% of the total variance was explained by differences between participants.

Post-hoc analyses revealed significant differences between Large-left Non-Symbolic trials and Large-left Symbolic trials (t(4303) = -8.19, p < 0.001) (refer to S1 Table for comprehensive post-hoc tests). Moreover, mean deviations from the objective center of the line were significant only when participants responded to the Large-left Non-Symbolic trials (t(33) = -3.63, p < 0.001). Mean deviation in baseline trials was M = -0.59 mm, DS = 1.39 mm.

### Experiment 2

The results showed the flanker format (Symbolic *vs.* Non-Symbolic) had a significant impact on bisection (β = 0.50, SE = 0.06, t(4295.01945) = 8.13, p < 0.001), with larger deviations in the Symbolic format (M = 0.30 mm; DS = 1.63 mm) compared with the Non-Symbolic format (M = -0.14 mm; DS = 1.51 mm) (refer to Table A and Table B in S5 Table for comprehensive mean value). Instead, the orientations (Small-left *vs.* Large-left) did not have a significant effect on bisection (β = -0.02, SE = 0.06, t(4295.01074)= -0.40, p = 0.68). The interaction between flanker orientations and format was not significant (β = -0.13, SE = 0.08, t(4295.04315) = -1.46, p = 0.14) (refer to Fig 1b). Random intercept variance for participants

was $\sigma^2 = 2.02$ (SD = .45). The variance explained by fixed effects was $R^2m = .02$, while the total explained variance was $R^2c = .19$. The intraclass correlation coefficient (ICC) was .18, indicating that 18% of the total variance was explained by differences between participants.

Although the interaction was not significant an exploratory post-hoc analysis was performed. Post-hoc analyses showed significant differences between Large-left Non-Symbolic trials and Large-left Symbolic trials (t(4295) = -8.13, p < 0.001). There were significant differences between Small-left Non-Symbolic trials and Small-left Symbolic trials (t(4295) = -6.05, p < 0.001) (refer to S2 Table for comprehensive post-hoc tests).

Moreover, mean deviations from the objective center of the line were significant only when participants responded to Large-left Non Symbolic trials (t(33) = 2.78, p-value < 0.01). Mean deviation in baseline trials was M = -0.57 mm, DS = 1.37 mm.

## Experiment 3

The results showed that the flanker format (Symbolic *vs.* Non-Symbolic) had a significant impact on bisection ($\beta = 0.59$, SE = 0.06, t(4297.02035) = 9.72, p < 0.001), with larger deviations in the Non-Symbolic format trials (M = -0.47 mm; DS = 1.68 mm) compared with the Symbolic format trials (M = -0.14 mm; DS = 1.64 mm) (refer to Table A and Table B in S5 Table for comprehensive mean values). Similarly, the orientations (Small-left *vs.* Large-left) had a significant influence ($\beta = 0.31$, *SE* = 0.06, t(4297.00668) = 5.17, p < 0.001), with larger deviations in the Large-left orientation trials (M = -0.34 mm; DS = 1.69 mm) compared with the Small-left orientation trials (M = -0.27 mm; DS = 1.63 mm). Notably, an interaction effect was observed between the flanker orientations and format ($\beta = -0.50$, *SE* = 0.08, t(4297.00751) = -5.86, p < 0.001) (refer to Fig 1b). Random intercept variance for participants was $\sigma^2 = 1.99$ (SD = .74). The variance explained by fixed effects was $R^2m = .01$, while the total explained variance was $R^2c = .28$. The intraclass correlation coefficient (ICC) was .27, indicating that 27% of the total variance was explained by differences between participants.

Post-hoc analyses showed significant differences between Large-left Non-Symbolic trials and Large-left Symbolic trials (t(4297) = -9.72, p < 0.001) (refer to S3 Table for comprehensive post-hoc tests). Moreover, mean deviations from the objective center of the line were significant only when participants responded to Large-left Non-Symbolic trials (t(33) = -3.48, p = 0.001). Mean deviation in baseline trials was M = -0.41 mm, DS = 1.40 mm.

## Experiment 4

The results revealed that both flanker format (Symbolic *vs.* Non-Symbolic) ($\beta = 0.44$, SE = 0.06, t(4304.00773) = 6.87, p < 0.001) and orientations (Small-left *vs.* Large-left) ($\beta = -0.69$, SE = 0.09, t(4304.99971) = 7.03, p < 0.001) significantly influenced the bisection.

Specifically, there were larger deviations in the Non-Symbolic format trials (M = -0.22 mm; DS = 1.71 mm) compared with the Symbolic format trials (M = -0.12 mm; DS = 1.80 mm) and larger deviations in the Large-left orientations trials (M = -0.23 mm; DS = 1.76 mm) compared with the Small-left orientations trials (M = -0.12 mm; DS = 1.76 mm) (refer to Table A and Table B in S5 Table for comprehensive mean values). Moreover, a significant interaction effect was observed between the flanker orientations and format ($\beta = -0.69$, SE = 0.09, t(4304.00079) = -7.62, p < 0.001) (refer to Fig 1b). Random intercept variance for participants was $\sigma^2 = 2.23$ (SD = .86). The variance explained by fixed effects was $R^2m = .01$, while the total explained variance was $R^2c = .28$. The intraclass correlation coefficient (ICC) was .28, indicating that 28% of the total variance was explained by differences between participants.

Post-hoc analyses revealed significant difference between Large-left Non-Symbolic and Large-left Symbolic (t(4304) = -6.87, p = 0.001) and between Small-left Non-Symbolic and Small-left Symbolic (t(4304)= 3.90, p = 0.0006) (refer to S4 Table for comprehensive post-hoc tests).

Moreover, mean deviations from the objective center of the line were significant only when participants responded to Large Non-Symbolic trials (t(33) = -2.60, p = 0.01). Mean deviation in baseline trials was M = -0.81 mm, DS = 1.33 mm.

## Discussion

In the 4 experiments presented here, a set of number line bisection tasks was used to investigate factors (perceptual, attentional or conceptual) that influence the processing of spatial extension. Specifically, our study focused on the line bisection task by exploring how the properties of two flankers positioned at both ends of each line modulated participants' performance. Crucially, the physical properties and the numerical value of the flankers were manipulated in an effort to separately explore the relative weight of each factor over the orientation of the MNL. Drawing from earlier research, we expected a divergent and opposite bisection in tasks where the small stimulus was on the left compared to tasks where the small stimulus was on the right [7,15,16,18,21].

Moreover, numerical flankers were displayed using both Symbolic (i.e., Arabic numbers) and Non-Symbolic stimuli (i.e., dots). Based on previous studies arguing for the existence of a common system for Symbolic and Non-Symbolic space-number representations [i.e., 7,8,41] we expected that Symbolic stimuli would elicit responses similar to the Non-Symbolic stimuli in this task.

In what follows, we summarize the main results obtained in each experiment. Thereafter, we provide comprehensive interpretations of the phenomena observed.

In Experiment 1, where numerical information was unevenly distributed across flankers (2–8 or 8–2), based on previous evidence, we expected similar performance regardless of format, with a tendency to bisect towards the flanker stimuli with the largest amount. Contrary to this prediction, the effect of the larger number was not consistently observed across conditions, except in the Non-Symbolic format and with the Larger-Left orientation.

In Experiment 2, where flankers were identical and symmetrically positioned, we expected that the perceptual factors thus controlled would lead to more accurate identification of the central dot or alternatively, a slight leftward bias, consistent with the manifestation of the pseudoneglect phenomenon [25]. Here, once again, the results were only partially consistent with the initial hypothesis. While the expected leftward bias was found in the Non-Symbolic format, rightward deviations were observed in the Symbolic format.

In Experiments 3 and 4, where the flankers were numerically identical but differed in perceptual salience, we expected a bias towards the most salient stimuli, independently of stimuli format. Larger deviations were recorded in the Non-symbolic compared to the Symbolic format, indicating a greater influence of perceptual features over the former. Thus, the abstract numerical information extracted from the flanker seems subordinated in this context.

Taken together, the findings indicate a main effect of the Format, supporting the view that distinct mechanisms might operate in Symbolic and Non-Symbolic processing [29–36]. Thus, the present study extends previous literature by providing unprecedented evidence for separate processing in bisection tasks [for a review see 42]. Here, differently from previous investigations with the line bisection [i.e., 7] participants systematically tended to show significantly larger leftward biases in the Non-Symbolic format compared to the Symbolic format.

The behavior in the Non-Symbolic format is compatible with pseudoneglect [25], a slight leftward bias caused by the anatomical organization of attention-related neural networks in the brain, in particular by the right hemisphere dominant activity in allocating spatial attention [43]. The Symbolic format, in contrast, evidenced a rightward bias or, when showing a leftward bias, it was less pronounced than in the Non-Symbolic format. This suggests the involvement of additional factors besides pseudoneglect.

To identify these processes, one should take into account the link between attention deployment and subjective line extent. Research shows that line length may appear longer depending on numerous factors such as the scanning direction over the line, lateralized cueing [25], and to endowment effects such as the mental representation of numbers [16].

Studies indicate that saliency of the flankers may also influence the outcome. For example, Arabic digits increase the saliency of the number magnitude representation in number line bisection tasks via the semantic route [44].

Most relevant for the present findings is the observation that symbolic processing is strongly left-lateralized in the intraparietal sulcus [45] and it activates the motor and visual areas in the brain more strongly than baseline line bisection tasks without digits [46]. This tentatively accounts for the rightward shift in the deviations in the Symbolic format compared to the Non-Symbolic format. We observed those differences, however, only under certain circumstances, as reflected by the interactions between numerical format and orientation. Specifically, the data indicate that the left tendency in the Non-Symbolic format was most prominent when the numerically larger flankers were on the left (i.e., Large-left orientation) (Experiment 1) or when the numerically identical flankers with more elements grouped in the proximity of the line were located in the left edge (Experiments 3 and 4), but not when the flankers were numerically equivalent and symmetrical (Experiment 2). By comparing these effects across studies, it seems likely that the imbalance between two flankers (either numerical or perceptual) contributes at directing participants' attention towards the most salient or larger stimuli.

It follows that, because our participants in the Non-Symbolic format tended to pay attention first to the most salient flanker, their performance in the task varied according to the two orientations presented. It is thus possible that participants adopt a bisection strategy that systematically scans from left to right, linked to cultural bias (e.g., reading direction), but when cue oriented in a different direction (due to saliency disparities of the flankers) they adopt a situational behavior, displaying an opposite bisection pattern.

Indeed, the leftward responses consequent of pseudoneglect were enhanced when the larger flankers were on the left (i.e., Large-left orientation). This is likely due to the deployment of attention to the most prominent flanker and the consequent left-to-right scanning over the line. Conversely, when the larger flankers were on the right (i.e., Small-left orientation) and attention was directed accordingly (inducing right-to-left scanning), and the pseudoneglect phenomenon was counteracted, yielding responses closer to the midpoint.

Noticeably, a perceptual imbalance - which even numerically comparable yet asymmetrically distributed arrays can elicit (Experiments 3 & 4) - seems to be at the roots of these effects. Congruently, in the case of perceptual balance (Experiment 2), in which no scanning biases are expected, comparable responses in the two orientations were observed.

The results in the Symbolic format evidence - in agreement with previous research [15,20] - that number and magnitude representations can further modulate the attentional and perceptual effects found for the Non-Symbolic format. The results in the Symbolic format, specifically in the Large-left orientation, suggest that participants' attention to the flankers followed the standard numerical order (from smaller to larger). That is, in the case of Large-left orientation trials, they seem to attend the right (Small quantity) first. This prompts rightward deviations that minimize any effects of pseudoneglect, and accounts for the negligible deviations from the veridical center that appear, on average, in these trials. This overall pattern in Large-left orientation trials was found when the Symbols differed - and thus could be ordered - on the basis of number (Experiment 1) or physical size (Experiments 3 & 4), but not when they were identical in size and number (Experiment 2). The extensive rightward deviations in the Symbolic format in Experiment 2 reveal how, in the absence of perceptual or numerical imbalance, the mere effects of stimulus format prevail.

It is important to note that this study has some limitations which provide avenues for future research. A first limitation concerns the effect sizes, which appear to be modest and therefore could be influenced by the accuracy of the measurement itself. Moreover, we cannot exclude that the decision to employ a pencil-and-paper task, although grounded in the field's most standard modality, might have influenced the findings. Previous studies have shown that performance on line bisection, while highly correlated, can differ across modalities [i.e., computerized vs. paper-and-pencil; 47]. This is possibly due to factors such as motor demands, effector or response modality, feedback or hand position adjustments, and to human measurement errors which vary between the two formats.

Recent efforts within the wider neuropsychology field aim to identify complementary methods, such as computerized or virtual reality-based assessments, to enhance sensitivity [48], or automatic measurements to increase accuracy [22]. Even

though a technology-based gold standard for tasks such as line bisection has yet to be established, subsequent studies could investigate the effects of using a computer or tablet in this context. A further limitation concerns the manipulation of Non-Symbolic numerosities. It would be beneficial in the future studies to investigate how participants bisect the line when non-numerical variables such as the quantity and area of dots are not congruent with the numerical representations. Finally, to corroborate our interpretations regarding scanning orientations, it would be useful to conduct new research with the use of the eye tracker. This would monitor participants' eye movements during the experiment, providing a more precise understanding of their visual exploration strategies. Eye tracking could also reveal whether participants explore Symbolic and Non-Symbolic formats differently, providing direct evidence on the role of Symbolic numbers in managing attentional resources.

To conclude, the results of these experiments showed distinct effects in line bisection estimations when Symbolic and Non-Symbolic numerosities are used. Specifically, while performance in both Non-Symbolic and Symbolic stimuli is influenced by participants' perceptual and visuo-spatial attentional biases, only responses in the Symbolic format seem influenced by an organized mental representation of numbers. Therefore, these findings support recent evidence suggesting independent representations for Symbolic and Non-Symbolic numerals [29,33,42,49,50], and with models [51] assuming that the association between numbers and space depends on automatic links but also task-specific features.

## Supporting information

**S1 Table. Post-Hoc results in Linear Mixed-Effects Models Results between flankers Numerosities and Format in Experiment 1.**
(DOCX)

**S2 Table. Post-Hoc results in Linear Mixed-Effects Models Results between flankers Numerosities and Format in Experiment 2.**
(DOCX)

**S3 Table. Post-Hoc results in Linear Mixed-Effects Models Results between flankers Numerosities and Format in Experiment 3.**
(DOCX)

**S4 Table. Post-Hoc results in Linear Mixed-Effects Models Results between flankers Numerosities and Format in Experiment 4.**
(DOCX)

**S5 Table. A. Mean and Standard Deviations of each Experiment combining format and orientation. B.** Mean and Standard Deviations of each Experiment by Format and Orientation.(DOCX)

**S6 Table. T-test baseline * (format+condition).** T-test was performed to compare baseline with format and orientation combined, based on the subjects' means.
(DOCX)

**S7 File. ANOVA (format x orientation x line length).**
(DOCX)

## Acknowledgments

We wish to thank Paola Cazzol and Paria Ahookhosh for their contribution in recruiting participants and data coding of the current paper.

## Author contributions

**Conceptualization:** Silvia Benavides-Varela, Rosa Rugani.

**Data curation:** Annamaria Porru, Lucia Ronconi.

**Formal analysis:** Annamaria Porru, Lucia Ronconi.

**Investigation:** Annamaria Porru, Silvia Benavides-Varela, Rosa Rugani.

**Methodology:** Silvia Benavides-Varela, Rosa Rugani.

**Visualization:** Annamaria Porru, Silvia Benavides-Varela, Rosa Rugani.

**Writing – original draft:** Annamaria Porru, Silvia Benavides-Varela, Rosa Rugani.

**Writing – review & editing:** Lucia Ronconi, Daniela Lucangeli, Lucia Regolin, Silvia Benavides-Varela, Rosa Rugani.

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
