## [Decision Letter · Decision Letter 0]

23 Jan 2025

PONE-D-24-53823Symbolic and Non-Symbolic numbers differently affect centre identification in a number-line bisection taskPLOS ONE

Dear Dr. Porru,

Thank you for submitting your manuscript to PLOS ONE. After careful consideration, we feel that it has merit but does not fully meet PLOS ONE’s publication criteria as it currently stands. Therefore, we invite you to submit a revised version of the manuscript that addresses the points raised during the review process.

<!--StartFragment I have now received comments from two reviewers with expertise in the field and also read your paper carefully myself. The reviewers’ comments are appended below. Although both Reviewers concur there is merit in your work, they also raise several conceptual and methodological concerns that prevent publication in its current form. I encourage you to resubmit a revised version of your manuscript where to address each of the points raised by the Reviewers, paying particular attention to align the purposes with the methods used as pointed out by Reviewer 1 and providing robust justification for the methods as emphasized by both Reviewers. You should also try to extend the Discussion section so to provide a more comprehensive interpretation of your results. 

We look forward to receiving your revised manuscript.

Kind regards,

Elisa Scerrati

Academic Editor

PLOS ONE

“European Union – Next Generation EU. for its support Prot.  PRIN 2022 PRIN - 202254RHRT to R.R., and PRIN 2022 PNRR - P2022TKY7B to S.B-V.”

Reviewers' comments:

Reviewer's Responses to Questions

**Comments to the Author**

1. Is the manuscript technically sound, and do the data support the conclusions?

Reviewer #1: No

Reviewer #2: Yes

2. Has the statistical analysis been performed appropriately and rigorously? 

Reviewer #1: No

Reviewer #2: Yes

3. Have the authors made all data underlying the findings in their manuscript fully available?

Reviewer #1: Yes

Reviewer #2: Yes

4. Is the manuscript presented in an intelligible fashion and written in standard English?

Reviewer #1: Yes

Reviewer #2: Yes

5. Review Comments to the Author

Reviewer #1: Summary

The study evaluated effects of task-irrelevant numerical flanker features while 136 neurotypical adults bisected lines in 4 experiments. Experiment 1 (N=36) compared flanker pair 8-2 in symbolic and non-symbolic formats in two orders. Left bias was largest for large left non-symbolic condition. Experiment 2 (N=36) studied perceptually symmetrical symbolic and non-symbolic flankers. Bias was positive for symbolic and negative for non-symbolic flankers, and again largest for large left non-symbolic condition. Experiment 3 (N=36) studied font and spacing asymmetries for symbolic and non-symbolic 8. Again, largest left bias obtained for large left non-symbolic condition. The final Experiment 4 (N=36) replicated font and spacing asymmetries for symbolic and non-symbolic 2. Again, largest left bias obtained for large left non-symbolic condition. The results are taken to speculate about format-specific representation of number-space associations.

Evaluation

Although the experiment examined interesting perceptual conditions and wisely included a baseline assessment, I have conceptual and methodological problems with the current report that prevent me from recommending publication.

Main problems

0. Let me begin by providing relevant context that is missing from this manuscript. The line bisection task, naively adopted by neuropsychologists for efficient bed-side diagnostics (like other seemingly simple tasks, e.g., Corsi blocks), has long been studied to understand sensory, cognitive, and motor contributions to performance. Perceptual flankers were added to manipulate visual attention prior to bisection by demanding their verbal reporting. Ever since, perceptual and attentional contributions to flanked line bisection performance were discussed, but I have never encountered a discussion of conceptual representations in the context of flanker manipulations. This brings me to my first (conceptual) problem.

1. The objectives and methods seem misaligned. The main effect of number flanker format is taken to argue for distinct representations (e.g., line 327f, 402f). But the bisection task measures sensory/perceptual (and also motor) but not conceptual factors. This is evident in all four results (largest stimulus extent on the left creates largest left bias) and in the wider literature. Thinking of the bisector placement as the participant’s subjective balance point of a scale (the line), what can the effect of placing a picture of an elephant or ant vs the word elephant/ant reveal about the cognitive representation of elephants? Results will depend on the font size, picture size, etc. Consistent with this assessment, the present report identifies perceptual grouping effects, as did Fischer (1994). This study should be added to the review and the interpretation of results adjusted.

Fischer MH. Less attention and more perception in cued line bisection. Brain Cogn 1994;25:24–33

1. Turning to methodological issues, I believe statistics can be improved. Effects should be computed relative to the baseline bisection, not relative to zero. ANOVA should (also) be performed on subject means (as was the t-test, i.e. with 33 degrees of freedom). Importantly, the factor line length (cf line 211) should be included to reduce error variance and increase effect sizes (over which the authors complain in line 388).

2. The literature review is suboptimal because of unclear formulations (see my many minor comments below). Readers would benefit from stating explicitly the two main effect patterns reported in this literature, namely a tendency to orient towards larger surface flankers and towards larger symbolic magnitudes.

3. Hypotheses should also be stated more clearly. For Experiment 1, the expectation is not “similar results” but “similar bias towards the larger number”. For Experiment 2 the hypothesis is not accurate bisection because of the extensive evidence for pseudo-neglect, which should be explained better and explicitly credited to Jewell & Mc Court (2000) in line 82 as part of the literature review, and again in lines 130 and 144.

4. I was surprised to see the study was conducted on paper (136 participants x 132 trials with single lines per page = 17,952 sheets of paper!). Since several of the experiments reviewed as background were conducted on computer displays, the authors should justify their decision and discuss whether the medium of testing might have contributed to the alleged heterogeneity of results (in the Intro) as well as to their own findings (in the Discussion). Luh et al. (1995) is an example of a comparative bisection study.

Luh, K. E. (1995). Line bisection and perceptual asymmetries in normal individuals: What you see is not what you get. Neuropsychology, 9(4), 435–448. https://doi.org/10.1037/0894-4105.9.4.435

5. The authors feel compelled to invoke left-lateralized cortical symbol processing (lines 341 ff) to account for right-bias with symbolic numbers. First, this incomplete argument should be augmented by explaining to readers (a) the cross-lateralization of visual processing, (b) the relation between hemispheric activity and attention deployment, and (c) the link between attention deployment and subjective line extent. Secondly, the “right bias” is only a relative bias, depending on baseline performance (which incidentally depends on line length). The same relativity holds for “left bias”, which is on top of the hemi-neglect bias. Awareness of this compound nature of bisection effects should inform the interpretation.

Minor issues

Line 68: the subitizing range is irrelevant for symbolic number processing, so the authors need to clarify that the study used non-symbolic numbers. Moreover, 9 is not within the subitizing range.

Line 80: the authors should describe the number position effect in full to be more useful for the readers. In general, the entire description can be improved (e.g., line 83: near or far from what? Line 81: what is a lateral disparity?)

Line 99: what is the comparison task?

Line 110: closer to what?

Line 146: the expression “denser in the line proximity” contains two aspects (density and proximity) – please clarify.

Line 166: Was handedness self-reported or measured?

Line 182: Clarify whether the baseline was always the first four trials (as implied by the word “phase”) or whether these were within the four blocks.

Line 209: what is “gsm” and which part of a stimulus was printed in white (assuming the pages were white)?

Line 201: Lines were centered on the page –inviting use of the page as spatial reference and potentially reducing experimentally induced effects. The authors might want to acknowledge this.

Line 220: this is the only mention of an “invisible square” – what was its purpose etc? I assume the answer will also clarify in what sense flanker digits for horizontally displayed lines can be horizontally centered” (would this not mean they appear above the line?).

Line 262: Report baseline mean and add the unit mm to all means.

Line 284: this sentence is incomplete.

Reviewer #2: The manuscript (PONE-D-24-53823) presents a Research Article treating four line-bisection experiments which feature different kinds of numerical flankers, either symbolic or non-symbolic in format. Stronger leftward biases on line bisection are found in the non-symbolic format, leading the authors to conclude that processes akin to pseudoneglect might account for this; whereas the rightward biases encountered in the symbolic format are said to reflect an organized representation of numbers.

While the manuscript is well written, the design carefully thought and statistical analyses adequately done, I have some major concerns that should in my opinion be addressed before proceeding with the publication process. Therefore, my suggestion is to carry out a major revision.

1. I have some doubts about the methodology – detailed below – which do not necessarily question the soundness of the study but that should be addressed, nonetheless.

2. The manuscript is overly brief when detailing the experimental manipulations and their operational relevance to test the study’s hypothesis.

3. I did not see a great effort in discussing the two main results, which crucially differ depending on the flanker’s format (left/right-ward biases for symbolic/non-symbolic), jointly. In the end, each of them is traced back to theoretical explanations that are not novel (i.e., pseudoneglect and scanning habits). While this is not bad in principle, I think the great work carried out to design this study deserves to be discussed in little more depth.

Detailed comments

• Line 61-62: this statement is quite generic and seems unnecessary.

• Lines 65 – 66: what do the leftward and rightward biases substantiate in?

• Line 80: the results from this experiment are not clear. It is particularly not clear what “the position of the strings on the sheet” means.

• Lines 76 – 91: I do not think that dedicating an entire paragraph to one single study is justified here, especially because it comprises many studies, the results of which cannot be addressed in detail for obvious reasons. I suggest to briefly outline the rationale behind the study and skipping to the general conclusion as at lines 89 – 91.

• Lines 110 – 111: the relationship between Gebuis and Gevers’ and the claim that introduces the paragraph (“distinction between symbolic and non-symbolic numbers’ representations” – line 106) is not clear.

• Lines 185 – 187: “Moreover … of the line”. This is not clear. What is the most “prominent” flanker? Why would it be placed on the left side of the line only? Please articulate.

• Figure 1 is in low resolution; moreover, it is overly concise. I think splitting it into two figures would make more sense.

• Lines 221 – 236: referencing to the figure and panel would help. Please also add a title for the very first column in figure 1 panel a – in which are the definitions Baseline, Small-left, Large-left…

• Lines 221, 223, 226 and any other occurrence: Please be consistent when referencing to the flankers: either use flankers, flanker stimuli or stimuli.

• Lines 223 – 224: “The stimuli “2-2” were arbitrarily considered the Small-left orientation and the “8-8” were considered the Large-left orientation” I do not understand this arbitrary choice, more so because I did not understand the rationale for this manipulation as explained at lines 185 – 187.

• Lines 332 – 335 “The behaviour in the Non-symbolic format is compatible with pseudoneglect” Was the leftward bias at all present in the symbolic format, though less intense? If that is the case, then I suggest rephrasing.

• Lines 357 – 364: I found this paragraph quite confusing, particularly so because there is an overlap between the terminology used by the authors (i.e., “scanning from left-to-right”) and what we usually refer to when talking about cultural-linguistic scanning. The authors should clarify if they are treating a cultural habit or a situational behaviour. In the latter case, I still believe the cultural bias for that should at least be acknowledged.

• Lines 368 – 369: “a right-to-left direction bias possibly counteracts the pseudoneglect phenomenon, yielding responses closer to the midpoint”. This was addressed a little too superficially. What would motivate such a bias?

General comments

• Procedure – this part could go into little more detail. How long were the lines? How big were the flankers? What were the numerosity of the flankers (both for symbolic and non-symbolic) How long on average did it take to hand the single sheets? Why was a pencil-and-paper version choosing instead of a digital one?

• *Edit* - while continuing reading, I realized the details missing in the procedure are actually in the Stimuli and Materials. I am leaving my prior comment because readers could get confused like me for not being able to read these details before the procedure. Please consider switching the two sections (Stimuli first – procedure second)

• I think a further justification for the paper-and-pencil methodology other than citing De Hevia and Spelke is needed. This choice leaves me puzzled as I believe precision in collecting and coding data has been undermined by this choice. How have potential biases or human errors been prevented? For instance, I am worried that letting participants carry the bisection manually could have let them adjust their hand position as to securely bisect it more or less at the same spot. Please elaborate on this.

• I would have appreciated a very brief introduction and/or discussion section for each experiment, summarizing the conditions in focus and the relevance of the results from each experiment for the hypothesis that informed its design. It would help focus the relevance of each experiment for the general research question.

Typos and minor concerns

• Line 73: “systematically bias”

• Line 192: You used “numerosity 8” and “numerosity two”. Please homologate the format.

• Line 284: “Moreover, deviations from the objective center of the line.” I guess this sentence is incomplete.

6. PLOS authors have the option to publish the peer review history of their article (what does this mean? ). If published, this will include your full peer review and any attached files.

**Do you want your identity to be public for this peer review?** For information about this choice, including consent withdrawal, please see our Privacy Policy .

Reviewer #1: No

Reviewer #2: No

---

## [Author Response · Author response to Decision Letter 1]

10 Mar 2025

March 10th, 2025

Elisa Scerrati, Ph.D.  

Academic Editor

PLOS ONE

Dear Prof. Scerrati,

Thank you very much for your consideration of our manuscript and for the opportunity to submit a revised version of it. We appreciate the time and effort that the reviewers have dedicated to providing valuable comments. We have engaged in the revisions based on the two reviewers, with particular attention to providing justification to the methods and extending the introduction and discussion, as suggested. Moreover, we have directly addressed all the concerns raised and included a point-by-point response of each below. All the corresponding changes have been highlighted in blue in the manuscript.

We believe that this revision has substantially clarified and improved our manuscript. We hope that you and the reviewers agree and consider accepting this manuscript for publication in PLOS ONE

Yours sincerely,

The authors

RESPONSE TO REVIEWER 1

0. Let me begin by providing relevant context that is missing from this manuscript.

The line bisection task, naively adopted by neuropsychologists for efficient bed-side diagnostics (like other seemingly simple tasks, e.g., Corsi blocks), has long been studied to understand sensory, cognitive, and motor contributions to performance. Perceptual flankers were added to manipulate visual attention prior to bisection by demanding their verbal reporting.

Ever since, perceptual and attentional contributions to flanked line bisection performance were discussed, but I have never encountered a discussion of conceptual representations in the context of flanker manipulations. This brings me to my first (conceptual) problem.

1. The objectives and methods seem misaligned. The main effect of the number flanker format is taken to argue for distinct representations (e.g., line 327f, 402f). But the bisection task measures sensory/perceptual (and also motor) but not conceptual factors. This is evident in all four results (largest stimulus extent on the left creates largest left bias) and in the wider literature. Thinking of the bisector placement as the participant’s subjective balance point of a scale (the line), what can the effect of placing a picture of an elephant or ant vs the word elephant/ant reveal about the cognitive representation of elephants? Results will depend on the font size, picture size, etc. Consistent with this assessment, the present report identifies perceptual grouping effects, as did Fischer (1994). This study should be added to the review and the interpretation of results adjusted.

Fischer MH. Less attention and more perception in cued line bisection. Brain Cogn 1994;25:24–33

We thank the Reviewer for this insightful comment, which captures the essence of the study. Although the line bisection task has been widely used in studies of cognitive psychology and neuropsychology, whether biases result from perceptual, attentional or conceptual factors is still a matter of controversy, particularly in the field of numerical cognition. Crucially, previous studies found that, even in the absence of explicit cueing of the flankers or requests for verbal reports of them, healthy adult participants would manifest a “conceptual” representation of flankers (i.e. magnitudes) compatible with the organization over a mental number line. The authors also found that this representation is abstract, insofar as it extends to symbolic and non-symbolic formats. The present study attempts to replicate such findings, and to extend them by manipulating perceptual cues in the original stimuli. As it turns out, these manipulations converge on the relevance of perceptual cues more than on its abstract representations.

In the revised version of the manuscript, we have exposed these controversies in the introduction to provide the reader with a more comprehensive view of the literature (p 3; lines 68). In doing that, the study by Fischer (1994) becomes relevant, along with other studies. We thus thank the reviewer for this suggestion.

1. Turning to methodological issues, I believe statistics can be improved. Effects should be computed relative to the baseline bisection not relative to zero. ANOVA should (also) be performed on subject means (as was the t-test, i.e. with 33 degrees of freedom). Importantly, the factor line length (cf line 211) should be included to reduce error variance and increase effect sizes (over which the authors complain in line 388).

We thank the reviewer for these suggestions.

Let us first clarify that most of our methodological decisions were made with the aim of providing a common framework to make meaningful comparisons between ours and previous studies. The decision to report the deviations compared to the objective midline is one of them. To the best of our knowledge, there is no study in the literature of number processing that reports and discusses the effects in relation to the baseline (subjective performance). Conversely, all of them report the results relative to zero (objective midline). Still, we acknowledge that small deviations in marking the midpoint may result from motor variability across participants. Thus, the requested comparisons are now included in the Supplementary Materials.

Similarly, the factor line length -agreeably relevant for the task (but see contradictory results in the metaanalysis of Jewell & McCourt, 2000)- was manipulated but not analyzed in studies in which the present study was built upon (i.e., DeHevia & Spelke, 2009). However, acknowledging the Reviewer’s suggestion, we have included line length as a factor in an ANOVA performed on subject means, as requested. The results of this new set of analyses are also included in the Supporting Information (p 33-35).

2. The literature review is suboptimal because of unclear formulations (see my many minor comments below). Readers would benefit from stating explicitly the two main effect patterns reported in this literature, namely a tendency to orient towards larger surface flankers and towards larger symbolic magnitudes.

We thank the reviewer for allowing us to clarify these points. We have reformulated the text in the introduction and explicitly inform the readers about the two most relevant and originally expected patterns (p 3, lines 68-70).

3. Hypotheses should also be stated more clearly. For Experiment 1, the expectation is not “similar results” but “similar bias towards the larger number”. For Experiment 2 the hypothesis is not accurate bisection because of the extensive evidence for pseudo-neglect, which should be explained better and explicitly credited to Jewell & Mc Court (2000) in line 82 as part of the literature review, and again in lines 130 and 144.

The new version of the manuscript implements a reformulation in an attempt to make the research hypotheses more explicit (p 4, lines 25), as suggested by you and by Reviewer 2.

A definition of the pseudoneglect and corresponding reference to Jewell & McCourt (2000) has been also included.

4. I was surprised to see the study was conducted on paper (136 participants x 132 trials with single lines per page = 17,952 sheets of paper!). Since several of the experiments reviewed as background were conducted on computer displays, the authors should justify their decision and discuss whether the medium of testing might have contributed to the alleged heterogeneity of results (in the Intro) as well as to their own findings (in the Discussion). Luh et al. (1995) is an example of a comparative bisection study.

Luh, K. E. (1995). Line bisection and perceptual asymmetries in normal individuals: What you see is not what you get. Neuropsychology, 9(4), 435–448. https://doi.org/10.1037/0894-4105.9.4.435

As explained above, the methodological implementations of the current studies were designed to be equivalent to previous studies evaluating the effects of numerical flankers on the line bisection. To the best of our knowledge, there are eight studies (out of ten) that used paper-and-pencil in this context. We thus deputed this modality not only as the most conventional but also comparable with previous experimental manipulations. This information has been included in the text (p. 10, lines 240,241). This is not to deny that other implementations could be more precise. Possible limitations of this modality have been now acknowledged in the discussion (p. 19-20, lines 464-468).

We also thank the reviewer for pointing out the article by Luh (1995), which was now included in the discussion regarding possible implications of this methodological choice.

5. The authors feel compelled to invoke left-lateralized cortical symbol processing (lines 341 ff) to account for right-bias with symbolic numbers. First, this incomplete argument should be augmented by explaining to readers (a) the cross-lateralization of visual processing, (b) the relation between hemispheric activity and attention deployment, and (c) the link between attention deployment and subjective line extent. Secondly, the “right bias” is only a relative bias, depending on baseline performance (which incidentally depends on line length). The same relativity holds for “left bias”, which is on top of the hemi-neglect bias. Awareness of this compound nature of bisection effects should inform the interpretation.

We appreciate the reviewer’s request for additional information. A more extended explanation of the phenomenon has been now included in the discussion (p 17-18, lines 407-419).

Minor issues

Line 68: the subitizing range is irrelevant for symbolic number processing, so the authors need to clarify that the study used non-symbolic numbers. Moreover, 9 is not within the subitizing range.

Due to numerous modifications, this passage has been erased from the current version of the manuscript. We have modified the previous sentence accordingly (also based on Reviewer’s 2 suggestions) and integrated this information in a different section of the manuscript (p. 3,5). We do not mention the subitizing range in this new report (which originally referred to one of the flankers i.e., 2, not to both).

Line 80: the authors should describe the number position effect in full to be more useful for the readers. In general, the entire description can be improved (e.g., line 83: near or far from what?

The line position effect has been described in a more specific way (lines 71-79). Moreover we have attempted a more accurate description of the studies.

Line 81: what is a lateral disparity?

We meant illusory spatial extension. In line 95, we modified the phrase accordingly 127-130

Line 99: what is the comparison task?

This phrase has been erased in the current version of the manuscript.

Line 110: closer to what?

Closer to the midline point. We have changed it to “more accurately”, to clarify it. Line 116-118.

Line 146: the expression “denser in the line proximity” contains two aspects (density and proximity) – please clarify.

The phrase has been modified in the following way: “larger numbers or larger quantities closer to the line” (lines 147-157).

Line 166: Was handedness self-reported or measured?

It was self-reported. This has been clarified in the text, line 176.

Line 182: Clarify whether the baseline was always the first four trials (as implied by the word “phase”) or whether these were within the four blocks.

Yes. The reviewer is right. The baseline was always presented at the beginning of each experiment. We have added this information in line 247.

Line 209: what is “gsm” and which part of a stimulus was printed in white (assuming the pages were white)?

GSM stands for 'grams per square metre' and refers to the weight of the paper. The heavier the paper, the ticker and the higher the number of grams per square metre. The stimuli was black, printed on a white paper. This has been corrected in lines 185-186.

Line 201: Lines were centered on the page –inviting use of the page as spatial reference and potentially reducing experimentally induced effects. The authors might want to acknowledge this.

Thanks for this suggestion. The information has been added in lines 199-201.

Line 220: this is the only mention of an “invisible square” – what was its purpose etc? I assume the answer will also clarify in what sense flanker digits for horizontally displayed lines can be horizontally centered” (would this not mean they appear above the line?).

The stimuli were presented within an invisible square of 5cmx5cm that served as the framework for the accurate location of the digit across trials and conditions. The digit was vertically centered with respect to the square and horizontally aligned towards the line. The line was centered vertically, with the digits positioned equidistant from both ends. This information has been added in the text (line 195, 199-202).

Line 262: Report baseline mean and add the unit mm to all means.

We are indebted to the suggestion for prompting us to report the mean and standard deviation of the baseline for each experiment. We have also incorporated unit measures into all descriptive analyses.

Line 284: this sentence is incomplete.

Thanks. This was a typo. The sentence we deleted.

RESPONSE TO REVIEWER 2

The manuscript (PONE-D-24-53823) presents a Research Article treating four line-bisection experiments which feature different kinds of numerical flankers, either symbolic or non-symbolic in format. Stronger leftward biases on line bisection are found in the non-symbolic format, leading the authors to conclude that processes akin to pseudoneglect might account for this; whereas the rightward biases encountered in the symbolic format are said to reflect an organized representation of numbers.

While the manuscript is well written, the design carefully thought and statistical analyses adequately done, I have some major concerns that should in my opinion be addressed before proceeding with the publication process. Therefore, my suggestion is to carry out a major revision.

1. I have some doubts about the methodology – detailed below – which do not necessarily question the soundness of the study but that should be addressed, nonetheless.

The concerns regarding the methodology have been addressed below, following each specific comment (lines 62-70).

2. The manuscript is overly brief when detailing the experimental manipulations and their operational relevance to test the study’s hypothesis.

We thank the reviewer for this insight. The new version of the manuscript implements a reformulation in an attempt to make the research hypotheses more explicit and the experimental manipulations clearer (p 6-7).

3. I did not see a great effort in discussing the two main results, which crucially differ depending on the flanker’s format (left/right-ward biases for symbolic/non-symbolic), jointly. In the end, each of them is traced back to theoretical explanations that are not novel (i.e., pseudoneglect and scanning habits). While this is not bad in principle, I think the great work carried out to design this study deserves to be discussed in little more depth.

In this new version we have modified the discussion to highlight possible mechanisms behind the differences obtained with the two formats. Specifically, the influence of attentional deployment over the flankers depending on the saliency (and perceived line length) has been discussed, particularly relevant for the non-symbolic format. The interplay between the lateralization of the brain mechanisms involved in spatial attention and Arabic number processing has been also considered to account for opposite results in the symbolic format (lines 382-399).

Detailed comments

• Line 61-62: this statement is quite generic and seems unnecessary.

In this version of the manuscript we have attempted to develop better the connections between evolutionary number space associations and the mature forms of the mental number line (lines 58-64).

• Lines 65 – 66: what do the leftward and rightward biases substantiate in?

The results were explained in terms of the mental number line hypothesis which, in the new version of the manuscript, has been better explained in the introduction, before the results of Fischer's study (lines 71-78). 

---

## [Decision Letter · Decision Letter 1]

9 Apr 2025

Symbolic and Non-Symbolic numbers differently affect centre identification in a number-line bisection task

PONE-D-24-53823R1

Dear Dr. Porru, I am pleased to inform you that your manuscript has been judged scientifically suitable for publication and will be formally accepted for publication once it meets all outstanding technical requirements.

Kind regards,

Elisa Scerrati

Academic Editor

PLOS ONE

Additional Editor Comments (optional):

Reviewers' comments:

Reviewer's Responses to Questions

**Comments to the Author**

1. If the authors have adequately addressed your comments raised in a previous round of review and you feel that this manuscript is now acceptable for publication, you may indicate that here to bypass the “Comments to the Author” section, enter your conflict of interest statement in the “Confidential to Editor” section, and submit your "Accept" recommendation.

Reviewer #1: All comments have been addressed

Reviewer #2: All comments have been addressed

2. Is the manuscript technically sound, and do the data support the conclusions?

Reviewer #1: Yes

Reviewer #2: Yes

3. Has the statistical analysis been performed appropriately and rigorously? 

Reviewer #1: Yes

Reviewer #2: Yes

4. Have the authors made all data underlying the findings in their manuscript fully available?

Reviewer #1: Yes

Reviewer #2: Yes

5. Is the manuscript presented in an intelligible fashion and written in standard English?

Reviewer #1: Yes

Reviewer #2: Yes

6. Review Comments to the Author

Reviewer #1: The authors did a good job of addressing the reviews and have improved the ms accordingly. The revised Introduction is much more useful and appropriate, especially in clarifying the impact of conceptual influences on bisection. Hypotheses now flow more naturally from this review.

My two minor observations can easily be addressed during proof-reading.

L 397-9 seem contradictory, please check

L 501 The Fias reference lacks co-authors.

Reviewer #2: (No Response)

7. PLOS authors have the option to publish the peer review history of their article (what does this mean? ). If published, this will include your full peer review and any attached files.

**Do you want your identity to be public for this peer review?** For information about this choice, including consent withdrawal, please see our Privacy Policy .

Reviewer #1: No

Reviewer #2: No

---

## [Editor Report · Acceptance letter]

PONE-D-24-53823R1

PLOS ONE

Dear Dr. Porru,

I'm pleased to inform you that your manuscript has been deemed suitable for publication in PLOS ONE. Congratulations! Your manuscript is now being handed over to our production team.

Kind regards,

on behalf of

Dr. Elisa Scerrati

Academic Editor

PLOS ONE